# Iterative Refinement Graph Neural Network for Antibody Sequence-Structure Co-design

**Wengong Jin**[†]**, Jeremy Wohlwend**[*]**, Regina Barzilay**[*]**, Tommi Jaakkola**[*]
[†] Eric and Wendy Schmidt Center, Broad Institute of MIT and Harvard
[*] CSAIL, Massachusetts Institute of Technology
`{wengong,jwohlwend,regina,tommi}@csail.mit.edu`

## Abstract

Antibodies are versatile proteins that bind to pathogens like viruses and stimulate the adaptive immune system. The specificity of antibody binding is determined by complementarity-determining regions (CDRs) at the tips of these Y-shaped proteins. In this paper, we propose a generative model to automatically design the CDRs of antibodies with enhanced binding specificity or neutralization capabilities. Previous generative approaches formulate protein design as a structure-conditioned sequence generation task, assuming the desired 3D structure is given a priori. In contrast, we propose to co-design the sequence and 3D structure of CDRs as graphs. Our model unravels a sequence autoregressively while iteratively refining its predicted global structure. The inferred structure in turn guides subsequent residue choices. For efficiency, we model the conditional dependence between residues inside and outside of a CDR in a coarse-grained manner. Our method achieves superior log-likelihood on the test set and outperforms previous baselines in designing antibodies capable of neutralizing the SARS-CoV-2 virus[1].

## 1 Introduction

Monoclonal antibodies are increasingly adopted as therapeutics targeting a wide range of pathogens such as SARS-CoV-2 (Pinto et al., 2020). Since the binding specificity of these Y-shaped proteins is largely determined by their complementarity-determining regions (CDRs), the main goal of computational antibody design is to automate the creation of CDR subsequences with desired properties. This problem is particularly challenging due to the combinatorial search space of over $20^{60}$ possible CDR sequences and the small solution space which satisfies the desired constraints of binding affinity, stability, and synthesizability (Raybould et al., 2019).

There are three key modeling questions in CDR generation. The first is how to model the relation between a sequence and its underlying 3D structure. Generating sequences without the corresponding structure (Alley et al., 2019; Shin et al., 2021) can lead to sub-optimal performance (Ingraham et al., 2019), while generating from a predefined 3D structure (Ingraham et al., 2019) is not suitable for antibodies since the desired structure is rarely known a priori (Fischman & Ofran, 2018). Therefore, it is crucial to develop models that *co-design* the sequence and structure. The second question is how to model the conditional distribution of CDRs given the remainder of a sequence (*context*). Attention-based methods only model the conditional dependence at the sequence level, but the structural interaction between the CDR and its context is crucial for generation. The last question relates to the model's ability to optimize for various properties. Traditional physics-based methods (Lapidoth et al., 2015; Adolf-Bryfogle et al., 2018) focus on binding energy minimization, but in practice, our objective can be much more involved than binding energies (Liu et al., 2020).

In this paper, we represent a sequence-structure pair as a graph and formulate the co-design task as a graph generation problem. The graph representation allows us to model the conditional dependence between a CDR and its context at both the sequence and structure levels. Antibody graph generation poses unique challenges because the global structure is expected to change when new nodes are inserted. Previous autoregressive models (You et al., 2018; Gebauer et al., 2019) cannot

---

[1]Our code is available at `https://github.com/wengong-jin/RefineGNN`

modify a generated structure because they are trained under teacher forcing. Thus errors made in the previous steps can lead to a cascade of errors in subsequent generation steps. To address these problems, we propose a novel architecture which interleaves the generation of amino acid nodes with the prediction of 3D structures. The structure generation is based on an iterative refinement of a global graph rather than a sequential expansion of a partial graph with teacher forcing. Since the context sequence is long, we further introduce a coarsened graph representation by grouping nodes into blocks. We apply graph convolution at a coarser level to efficiently propagate the contextual information to the CDR residues. After pretraining our model on antibodies with known structures, we finetune it using a predefined property predictor to generate antibodies with specific properties.

We evaluate our method on three generation tasks, ranging from language modeling to SARS-CoV-2 neutralization optimization and antigen-binding antibody design. Our method is compared with a standard sequence model (Saka et al., 2021; Akbar et al., 2021) and a state-of-the-art graph generation method (You et al., 2018) tailored to antibodies. Our method not only achieves lower perplexity on test sequences but also outperforms previous baselines in property-guided antibody design tasks.

## 2    RELATED WORK

**Antibody/protein design.** Current methods for computational antibody design roughly fall into two categories. The first class is based on energy function optimization (Pantazes & Maranas, 2010; Li et al., 2014; Lapidoth et al., 2015; Adolf-Bryfogle et al., 2018), which use Monte Carlo simulation to iteratively modify a sequence and its structure until reaching a local energy minimum. Similar approaches are used in protein design (Leaver-Fay et al., 2011; Tischer et al., 2020). Nevertheless, these physics-based methods are computationally expensive (Ingraham et al., 2019) and our desired objective can be much more complicated than low binding energy (Liu et al., 2020).

The second class is based on generative models. For antibodies, they are mostly sequence-based (Alley et al., 2019; Shin et al., 2021; Saka et al., 2021; Akbar et al., 2021). For proteins, O'Connell et al. (2018); Ingraham et al. (2019); Strokach et al. (2020); Karimi et al. (2020); Cao et al. (2021) further developed models conditioned on a backbone structure or protein fold. Our model also seeks to incorporate 3D structure information for antibody generation. Since the best CDR structures are often unknown for new pathogens, we co-design sequences and structures for specific properties.

**Generative models for graphs.** Our work is related to autoregressive models for graph generation (You et al., 2018; Li et al., 2018; Liu et al., 2018; Liao et al., 2019; Jin et al., 2020a). In particular, Gebauer et al. (2019) developed G-SchNet for molecular graph and conformation co-design. Unlike our method, they generate edges sequentially and cannot modify a previously generated subgraph when new nodes arrive. While Graphite (Grover et al., 2019) also uses iterative refinement to predict the adjacency matrix of a graph, it assumes all the node labels are given and predicts edges only. In contrast, our work combines autoregressive models with iterative refinement to generate a full graph with node and edge labels, including node labels and coordinates.

**3D structure prediction.** Our approach is closely related to protein folding (Ingraham et al., 2018; Yang et al., 2020a; Baek et al., 2021; Jumper et al., 2021). Inputs to the state-of-the-art models like AlphaFold require a complete protein sequence, its multi-sequence alignment (MSA), and its template features. These models are not directly applicable because we need to predict the structure of an *incomplete* sequence and the MSA is not specified in advance.

Our iterative refinement model is also related to score matching methods for molecular conformation prediction (Shi et al., 2021) and diffusion-based methods for point clouds (Luo & Hu, 2021). These algorithms also iteratively refine a predicted 3D structure, but only for a complete molecule or point cloud. In contrast, our approach learns to predict the 3D structure for incomplete graphs and interleaves 3D structure refinement with graph generation.

## 3    ANTIBODY SEQUENCE AND STRUCTURE CO-DESIGN

**Overview.** The role of an antibody is to bind to an *antigen* (e.g. a virus), present it to the immune system, and stimulate an immune response. A subset of antibodies known as *neutralizing* antibodies not only bind to an antigen but can also suppress its activity. An antibody consists of a heavy chain and a light chain, each composed of one *variable domain* (VH/VL) and some constant domains. The

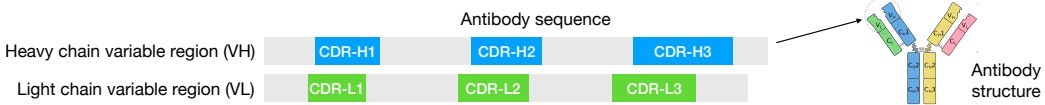

Figure 1: Schematic structure of an antibody (figure modified from Wikipedia).

variable domain is further divided into a *framework region* and three *complementarity determining regions* (CDRs). The three CDRs on the heavy chain are labeled as CDR-H1, CDR-H2, CDR-H3, each occupying a *contiguous* subsequence (Figure 1). As the most variable part of an antibody, CDRs are the main determinants of binding and neutralization (Abbas et al., 2014).

Following Shin et al. (2021); Akbar et al. (2021), we formulate antibody design as a CDR generation task, conditioned on the framework region. Specifically, we represent an antibody as a graph, which encodes both its sequence and 3D structure. We propose a new graph generation approach called RefineGNN and extend it to handle conditional generation given a fixed framework region. Lastly, we describe how to apply RefineGNN to property-guided optimization to design new antibodies with better neutralization properties. For simplicity, we focus on the generation of heavy chain CDRs, though our method can be easily extended to model light chains CDRs.

**Notations.** An antibody VH domain is represented as a sequence of amino acids $\boldsymbol{s} = \boldsymbol{s}_1 \boldsymbol{s}_2 \cdots \boldsymbol{s}_n$. Each token $\boldsymbol{s}_i$ in the sequence is called a *residue*, whose value can be either one of the 20 amino acids or a special token $\langle\text{MASK}\rangle$, meaning that its amino acid type is unknown and needs to be predicted. The VH sequence folds into a 3D structure and each residue $\boldsymbol{s}_i$ is labeled with three backbone coordinates: $\boldsymbol{x}_{i,\alpha}$ for its alpha carbon atom, $\boldsymbol{x}_{i,c}$ for its carbon atom, and $\boldsymbol{x}_{i,n}$ for its nitrogen atom.

### 3.1 GRAPH REPRESENTATION

We represent an antibody (VH) as a graph $\mathcal{G}(\boldsymbol{s}) = (\mathcal{V}, \mathcal{E})$ with node features $\mathcal{V} = \{\boldsymbol{v}_1, \cdots, \boldsymbol{v}_n\}$ and edge features $\mathcal{E} = \{\boldsymbol{e}_{ij}\}_{i \neq j}$. Each node feature $\boldsymbol{v}_i$ encodes three dihedral angles $(\phi_i, \psi_i, \omega_i)$ related to three backbone coordinates of residue $i$. For each residue $i$, we compute an orientation matrix $\boldsymbol{O}_i$ representing its local coordinate *frame* (Ingraham et al., 2019) (defined in the appendix). This allows us to compute edge features describing the spatial relationship between two residues $i$ and $j$:

$$\boldsymbol{e}_{ij} = \left( E_{\text{pos}}(i - j), \quad \text{RBF}(\|\boldsymbol{x}_{i,\alpha} - \boldsymbol{x}_{j,\alpha}\|), \quad \boldsymbol{O}_i^\top \frac{\boldsymbol{x}_{j,\alpha} - \boldsymbol{x}_{i,\alpha}}{\|\boldsymbol{x}_{i,,\alpha} - \boldsymbol{x}_{j,,\alpha}\|}, \quad \boldsymbol{q}(\boldsymbol{O}_i^\top \boldsymbol{O}_j) \right). \quad (1)$$

The edge feature $\boldsymbol{e}_{ij}$ contains four parts. The positional encoding $E_{\text{pos}}(i - j)$ encodes the relative distance between two residues in an antibody sequence. The second term $\text{RBF}(\cdot)$ is a *distance* encoding lifted into radial basis. The third term in $\boldsymbol{e}_{ij}$ is a *direction* encoding that corresponds to the relative direction of $\boldsymbol{x}_j$ in the local frame of residue $i$. The last term $\boldsymbol{q}(\boldsymbol{O}_i^\top \boldsymbol{O}_j)$ is the *orientation* encoding of the quaternion representation $\boldsymbol{q}(\cdot)$ of the spatial rotation matrix $\boldsymbol{O}_i^\top \boldsymbol{O}_j$. We only include edges in the $K$-nearest neighbors graph of $\mathcal{G}(\boldsymbol{s})$ with $K = 8$. For notation convenience, we use $\mathcal{G}$ as a shorthand for $\mathcal{G}(\boldsymbol{s})$ when there is no ambiguity.

### 3.2 ITERATIVE REFINEMENT GRAPH NEURAL NETWORK (REFINEGNN)

We propose to generate an antibody graph via an iterative refinement process. Let $\mathcal{G}^{(0)}$ be the initial guess of the true antibody graph. Each residue is initialized as a special token $\langle\text{MASK}\rangle$ and each edge $(i, j)$ is initialized to be of distance $3|i - j|$ since the average distance between consecutive residues is around three. The direction and orientation features are set to zero. In each generation step $t$, the model learns to revise a current antibody graph $\mathcal{G}^{(t)}$ and predict the label of the next residue $t + 1$. Specifically, it first encodes $\mathcal{G}^{(t)}$ with a message passing network (MPN) with parameter $\theta$

$$\{\boldsymbol{h}_1^{(t)}, \cdots, \boldsymbol{h}_n^{(t)}\} = \text{MPN}_\theta(\mathcal{G}^{(t)}), \quad (2)$$

where $\boldsymbol{h}_i^{(t)}$ is a learned representation of residue $i$ under the current graph $\mathcal{G}^{(t)}$. Our MPN consists of $L$ message passing layers with the following architecture

$$\boldsymbol{h}_i^{(t,l+1)} = \text{LayerNorm}\left( \sum_j \text{FFN}(\boldsymbol{h}_i^{(t,l)}, \boldsymbol{h}_j^{(t,l)}, E(\boldsymbol{s}_j), \boldsymbol{e}_{i,j}) \right), \quad 0 \leq l \leq L - 1, \quad (3)$$

where $\boldsymbol{h}_i^{(t,0)} = \boldsymbol{v}_i$ and $\boldsymbol{h}_i^{(t)} = \boldsymbol{h}_i^{(t,L)}$. FFN is a two-layer feed-forward network (FFN) with ReLU activation function. $E(\boldsymbol{s}_j)$ is a learned embedding of amino acid type $\boldsymbol{s}_j$. Based on the learned residue representations, we predict the amino acid type of the next residue $t+1$ (Figure 2A).

$$\boldsymbol{p}_{t+1} = \text{softmax}(\boldsymbol{W}_a \boldsymbol{h}_{t+1}^{(t)}) \tag{4}$$

This prediction gives us a new graph $\mathcal{G}^{(t+0.5)}$ with the same edges as $\mathcal{G}^{(t)}$ but the node label of $t+1$ is changed (Figure 2B). Next, we need to update the structure to accommodate the new residue $t+1$. To this end, we encode graph $\mathcal{G}^{(t+0.5)}$ by another MPN with a different parameter $\tilde{\theta}$ and predict the coordinate of all residues.

$$\{\boldsymbol{h}_1^{(t+0.5)}, \cdots, \boldsymbol{h}_n^{(t+0.5)}\} = \text{MPN}_{\tilde{\theta}}(\mathcal{G}^{(t+0.5)}) \tag{5}$$

$$\boldsymbol{x}_{i,e}^{(t+1)} = \boldsymbol{W}_x^e \boldsymbol{h}_i^{(t+0.5)}, \qquad 1 \leq i \leq n, e \in \{\alpha, c, n\}. \tag{6}$$

The new coordinates $\boldsymbol{x}_i^{(t+1)}$ define a new antibody graph $\mathcal{G}^{(t+1)}$ for the next iteration (Figure 2C). We explicitly realize the coordinates of each residue because we need to calculate the spatial edge features for $\mathcal{G}^{(t+1)}$. The structure prediction (coordinates $\boldsymbol{x}_i$) and sequence prediction (amino acid types $\boldsymbol{p}_{t+1}$) are carried out by two different MPNs, namely the *structure* network $\tilde{\theta}$ and *sequence* network $\theta$. This disentanglement allows the two networks to focus on two distinct tasks.

**Training.** During training, we only apply teacher forcing to the discrete amino acid type prediction. Specifically, in each generation step $t$, residues 1 to $t$ are set to their ground truth amino acid types $\boldsymbol{s}_1, \cdots, \boldsymbol{s}_t$, while all future residues $t+1, \cdots, n$ are set to a padding token. In contrast, the continuous structure prediction is carried out *without* teacher forcing. In each iteration, the model refines the *entire* structure predicted in the previous step and constructs a new $K$-nearest neighbors graph $\mathcal{G}^{(t+1)}$ of *all* residues based on the predicted coordinates $\{\boldsymbol{x}_{i,e}^{(t+1)} \mid 1 \leq i \leq n, e \in \{\alpha, c, n\}\}$.

**Loss function.** Our model remains rotation and translation invariant because the loss function is computed over pairwise distance and angles rather than coordinates. The loss function for antibody structure prediction consists of three parts.

- *Distance loss*: For each residue pair $i, j$, we compute its pairwise distance between the predicted alpha carbons $\boldsymbol{x}_{i,\alpha}^{(t)}, \boldsymbol{x}_{j,\alpha}^{(t)}$. We define the distance loss as the Huber loss between the predicted and true pairwise distances

$$\mathcal{L}_d^{(t)} = \sum_{i,j} \ell_{\text{huber}}(\|\boldsymbol{x}_{i,\alpha}^{(t)} - \boldsymbol{x}_{j,\alpha}^{(t)}\|^2, \|\boldsymbol{x}_{i,\alpha} - \boldsymbol{x}_{j,\alpha}\|^2), \tag{7}$$

where distance is squared to avoid the square root operation which causes numerical instability.

- *Dihedral angle loss*: For each residue, we calculate its dihedral angle $(\phi_i^{(t)}, \psi_i^{(t)}, \omega_i^{(t)})$ based on the predicted atom coordinates $\boldsymbol{x}_{i,\alpha}^{(t)}, \boldsymbol{x}_{i,c}^{(t)}, \boldsymbol{x}_{i,n}^{(t)}$ and $\boldsymbol{x}_{i+1,\alpha}^{(t)}, \boldsymbol{x}_{i+1,c}^{(t)}, \boldsymbol{x}_{i+1,n}^{(t)}$. We define the dihedral angle loss as the mean square error between the predicted and true dihedral angles

$$\mathcal{L}_a^{(t)} = \sum_i \sum_{a \in \{\phi, \psi, \omega\}} (\cos a_i^{(t)} - \cos a_i)^2 + (\sin a_i^{(t)} - \sin a_i)^2 \tag{8}$$

- $C_\alpha$ *angle loss*: We calculate angles $\gamma_i^{(t)}$ between two vectors $\boldsymbol{x}_{i-1,\alpha}^{(t)} - \boldsymbol{x}_{i,\alpha}^{(t)}$ and $\boldsymbol{x}_{i,\alpha}^{(t)} - \boldsymbol{x}_{i+1,\alpha}^{(t)}$ as well as dihedral angles $\beta_i^{(t)}$ between two planes defined by $\boldsymbol{x}_{i-2,\alpha}^{(t)}, \boldsymbol{x}_{i-1,\alpha}^{(t)}, \boldsymbol{x}_{i,\alpha}^{(t)}, \boldsymbol{x}_{i+1,\alpha}^{(t)}$.

$$\mathcal{L}_c^{(t)} = \sum_i (\cos \gamma_i^{(t)} - \cos \gamma_i)^2 + (\cos \beta_i^{(t)} - \cos \beta_i)^2 \tag{9}$$

In summary, the overall graph generation loss is defined as $\mathcal{L} = \mathcal{L}_{\text{seq}} + \mathcal{L}_{\text{struct}}$, where

$$\mathcal{L}_{\text{struct}} = \sum_t \mathcal{L}_d^{(t)} + \mathcal{L}_a^{(t)} + \mathcal{L}_c^{(t)} \qquad \mathcal{L}_{\text{seq}} = \sum_t \mathcal{L}_{ce}(\boldsymbol{p}_t, \boldsymbol{s}_t). \tag{10}$$

The sequence prediction loss $\mathcal{L}_{\text{seq}}$ is the cross entropy $\mathcal{L}_{ce}$ between predicted and true residue types.

### 3.3 CONDITIONAL GENERATION GIVEN THE FRAMEWORK REGION

The model architecture described so far is designed for unconditional generation — it generates an entire antibody graph without any constraints. In practice, we usually fix the framework region of an

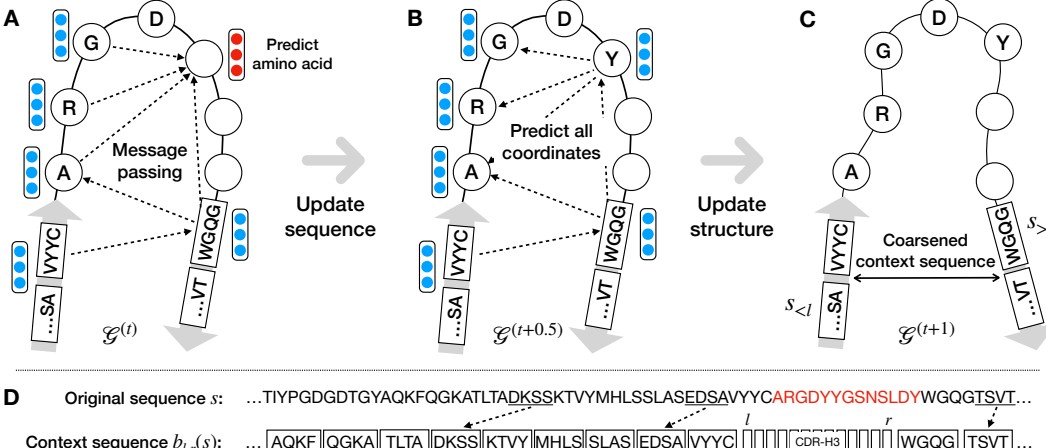

Figure 2: (A-C) One generation step of RefineGNN. Each circle represents a CDR residue and each square represents a residue block in a coarsened context sequence. (D) Sequence coarsening.

antibody and design the CDR sequence only. Therefore, we need to extend the model architecture to learn the conditional distribution $P(s'|s_{<l}, s_{>r})$, where $s_{<l} = s_1 \cdots s_{l-1}$ and $s_{>r} = s_{r+1} \cdots s_n$ are residues outside of the CDR $s_l \cdots s_r$.

**Conditioning via attention.** A simple extension of RefineGNN is to encode the non-CDR sequence using a recurrent neural network and propagate information to the CDR through an attention layer. To be specific, we first concatenate $s_{<l}$ and $s_{>r}$ into a *context* sequence $\tilde{s} = s_{<l} \oplus \langle \text{MASK} \rangle \cdots \langle \text{MASK} \rangle \oplus s_{>r}$, where $\oplus$ means string concatenation and $\langle \text{MASK} \rangle$ is repeated $n$ times. We then encode this context sequence by a Gated Recurrent Unit (GRU) (Cho et al., 2014) and modify the structure and sequence prediction step (Equation 4 and 6) as

$$\{c_1, \cdots, c_n\} = c_{1:n} = \text{GRU}(\tilde{s}) \tag{11}$$

$$p_{t+1} = \text{softmax}\big(W_a h_{t+1}^{(t)} + U_a^\top \text{attention}(c_{1:n}, h_{t+1}^{(t)})\big) \tag{12}$$

$$x_{i,e}^{(t+1)} = W_x^e h_i^{(t+0.5)} + U_x^{e\top} \text{attention}(c_{1:n}, h_i^{(t+0.5)}) \tag{13}$$

**Multi-resolution modeling.** The attention-based approach alone is not sufficient because it does not model the structure of the context sequence, thus ignoring how its residues *structurally* interact with the CDR's. While this information is not available for new antibodies at test time, we can learn to predict this interaction using antibodies in the training set with known structures.

A naive solution is to iteratively refine the entire antibody structure (more than 100 residues) while generating CDR residues. This approach is computationally expensive because we need to recompute the MPN encoding for all residues in each generation step. Importantly, we cannot predict the context residue coordinates at the outset and fix them because they need to be adjusted accordingly when the coordinates of CDR residues are updated in each generation step.

For computational efficiency, we propose a coarse-grained model that reduces the context sequence length by clustering it into *residue blocks*. Specifically, we construct a coarsened context sequence $b_{l,r}(s)$ by clustering every $b$ context residues into a block (Figure 2D). The new sequence $b_{l,r}(s)$ defines a coarsened graph $\mathcal{G}(b_{l,r}(s))$ over the residue blocks, whose edges are defined based on block coordinates. The coordinate of each block $x_{b_i,e}$ is defined as the mean coordinate of residues within the block. The embedding of each block $E(b_i)$ is the mean of its residue embeddings.

$$E(b_i) = \sum_{s_j \in b_i} E(s_j)/b, \qquad x_{b_i,e} = \sum_{s_j \in b_i} x_{j,e}/b, \qquad e \in \{\alpha, c, n\}. \tag{14}$$

Now we can apply RefineGNN to generate the CDR residues while iteratively refining the global graph $\mathcal{G}(b_{l,r}(s))$ by predicting the coordinates of all blocks. The only change is that the structure prediction loss is defined over block coordinates $x_{b_i,e}$. Lastly, we combine both the attention mechanism and coarse-grained modeling to keep both fine-grained and coarse-grained information. The decoding process of this conditional RefineGNN is illustrated in Algorithm 1.

---

**Algorithm 1** RefineGNN decoding

---

**Require:** Context sequence $\boldsymbol{s}_{<l}, \boldsymbol{s}_{>r}$
 1: Predict the CDR length $n$
 2: Coarsen the context sequence into $\boldsymbol{b}_{l,r}(\boldsymbol{s})$
 3: Construct the initial graph $\mathcal{G}^{(0)}$
 4: **for** $t = 0$ to $n - 1$ **do**
 5:     Encode $\mathcal{G}^{(t)}$ using the sequence MPN
 6:     Predict distribution of the next residue $\boldsymbol{p}_{t+1}$
 7:     Sample $\boldsymbol{s}_{t+1} \sim \text{categorical}(\boldsymbol{p}_{t+1})$
 8:     Encode $\mathcal{G}^{(t+0.5)}$ with the structure MPN
 9:     Predict all residue coordinates $\boldsymbol{x}_{i,e}^{(t+1)}$
10:     Update $\mathcal{G}^{(t+1)}$ using the new coordinates

---

**Algorithm 2** ITA-based sequence optimization

---

**Require:** A set of antibodies $\mathcal{D}$ to be optimized
**Require:** A neutralization predictor $f$.
**Require:** A set of neutralizing antibodies $Q$
 1: **for** each iteration **do**
 2:     Sample an antibody $\boldsymbol{s}$ from $\mathcal{D}$, remove its CDR and get a context sequence $\boldsymbol{b}_{l,r}(\boldsymbol{s})$
 3:     **for** $i = 1$ to $M$ **do**
 4:         Sample $\boldsymbol{s}_i' \sim P_\Theta(\boldsymbol{s}'|\boldsymbol{b}_{l,r}(\boldsymbol{s}))$
 5:         **if** $f(\boldsymbol{s}_i') > \max(f(\boldsymbol{s}), 0.5)$ **then**
 6:             $Q \leftarrow Q \cup \{\boldsymbol{s}_i'\}$
 7:     Sample a batch of new antibodies from $Q$
 8:     Update model parameter $\Theta$ by minimizing the sequence prediction loss $\mathcal{L}_{\text{seq}}$.

---

### 3.4 PROPERTY-GUIDED SEQUENCE OPTIMIZATION

Our ultimate goal is to generate new antibodies with desired properties such as neutralizing a particular virus. This task can be formulated as an optimization problem. Let $Y$ be a binary indicator variable for neutralization. Our goal is to learn a conditional generative model $P_\Theta(\boldsymbol{s}'|\boldsymbol{b}_{l,r}(\boldsymbol{s}))$ that maximizes the probability of neutralization for a training set of antibodies $\mathcal{D}$, i.e.

$$\sum_{\boldsymbol{s}\in\mathcal{D}} \log P(Y = 1|\boldsymbol{b}_{l,r}(\boldsymbol{s})) = \sum_{\boldsymbol{s}\in\mathcal{D}} \log \sum_{\boldsymbol{s}'} f(\boldsymbol{s}')P_\Theta(\boldsymbol{s}'|\boldsymbol{b}_{l,r}(\boldsymbol{s})) \tag{15}$$

where $f(\boldsymbol{s}')$ is a predictor for $P(Y = 1|\boldsymbol{s}')$. Assuming $f$ is given, this problem can be solved by iterative target augmentation (ITA) (Yang et al., 2020b). Before ITA optimization starts, we first pretrain our model on a set of real antibody structures to learn a prior distribution over CDR sequences and structures. In each ITA finetuning step, we first randomly sample a sequence $\boldsymbol{s}$ from $\mathcal{D}$, a set of antibodies whose CDRs need to be redesigned. Next, we generate $M$ new sequences given its context $\boldsymbol{b}_{l,r}(\boldsymbol{s})$. A generated sequence $\boldsymbol{s}_i'$ is added to our training set $Q$ if it is predicted as neutralizing. Initially, the training set $Q$ contains antibodies that are known to be neutralizing ($Y = 1$). Lastly, we sample a batch of neutralizing antibodies from $Q$ and update the model parameter by minimizing their sequence prediction loss $\mathcal{L}_{\text{seq}}$ (Eq.(10)). The structure prediction loss $\mathcal{L}_{\text{struct}}$ is excluded in ITA finetuning phase because the structure of a generated sequence is unknown.

## 4 EXPERIMENTS

**Setup.** We construct three evaluation setups to quantify the performance of our approach. Following standard practice in generative model evaluation, we first measure the perplexity of different models on new antibodies in a test set created based on sequence similarity split. We also measure structure prediction error by comparing generated and ground truth CDR structures recorded in the Structural Antibody Database (Dunbar et al., 2014). Results for this task are shown in section 4.1.

Second, we evaluate our method on an existing antibody design benchmark of 60 antibody-antigen complexes from Adolf-Bryfogle et al. (2018). The goal is to design the CDR-H3 of an antibody so that it binds to a given antigen. Results for this task are shown in section 4.2.

Lastly, we propose an antibody optimization task which aims to redesign CDR-H3 of antibodies in the Coronavirus Antibody Database (Raybould et al., 2021) to improve their neutralization against SARS-CoV-2. CDR-H3 design with a fixed framework is a common practice in the antibody engineering community (Adolf-Bryfogle et al., 2018; Liu et al., 2020). Following works in molecular design (Jin et al., 2020b), we use a predictor to evaluate the neutralization of generated antibodies since we cannot experimentally test them in wet labs. Results for this task are reported in section 4.3.

**Baselines.** We consider three baselines for comparison (details in the appendix). The first baseline is a sequence-based LSTM model used in Saka et al. (2021); Akbar et al. (2021). This model does not utilize any 3D structure information. It consists of an encoder that learns to encode a context sequence $\tilde{\boldsymbol{s}}$, a decoder that decodes a CDR sequence, and an attention layer connecting the two.

Table 1: *Left*: Language modeling results. We report perplexity (PPL) and root mean square deviation (RMSD) for each CDR in the heavy chain. *Right*: Results on the antigen-binding antibody design task. We report the amino acid recovery (AAR) for all methods.

| Model | CDR-H1 | | CDR-H2 | | CDR-H3 | | Model | AAR |
|---|---|---|---|---|---|---|---|---|
| | PPL | RMSD | PPL | RMSD | PPL | RMSD | RAbD | 28.53% |
| LSTM | 6.79 | - | 7.21 | - | 9.70 | - | LSTM | 22.53% |
| AR-GNN | 6.44 | 2.97 | 6.86 | 2.27 | 9.44 | 3.63 | AR-GNN | 23.86% |
| RefineGNN | **6.09** | **1.18** | **6.58** | **0.87** | **8.54** | **2.50** | RefineGNN | **34.14%** |

The second baseline is an autoregressive graph generation model (AR-GNN) whose architecture is similar to You et al. (2018); Jin et al. (2020b) but tailored for antibodies. AR-GNN generates an antibody graph residue by residue. In each step $t$, it first predicts the amino acid type of residue $t$ and then generates edges between $t$ and previous residues. Importantly, AR-GNN cannot modify a partially generated 3D structure of residues $s_1 \cdots s_{t-1}$ because it is trained by teacher forcing.

On the antigen-binding task, we include an additional physics-based baseline called RosettaAntibodyDesign (RAbD) (Adolf-Bryfogle et al., 2018). We apply their *de novo* design protocol composed of graft design followed by 250 iterations of sequence design and energy minimization. We cannot afford to run more iterations because it takes more than 10 hours per antibody. We also could not apply RAbD to the SARS-CoV-2 task because it requires 3D structures to be given. This information is unavailable for antibodies in CoVAbDab.

**Hyperparameters.** We performed hyperparameter tuning to find the best setting for each method. For RefineGNN, both its structure and sequence MPN have four message passing layers, with a hidden dimension of 256 and block size $b = 4$. All models are trained by the Adam optimizer with a learning rate of 0.001. More details are provided in the appendix.

## 4.1 LANGUAGE MODELING AND 3D STRUCTURE PREDICTION

**Data.** The Structural Antibody Database (SAbDab) consists of 4994 antibody structures renumbered according to the IMGT numbering scheme (Lefranc et al., 2003). To measure a model's ability to generalize to novel CDR sequences, we divide the heavy chains into training, validation, and test sets based on CDR cluster split. We illustrate our cluster split process using CDR-H3 as an example. First, we use MMseqs2 (Steinegger & Söding, 2017) to cluster all the CDR-H3 sequences. The sequence identity is calculated under the BLOSUM62 substitution matrix (Henikoff & Henikoff, 1992). Two antibodies are put into the same cluster if their CDR-H3 sequence identity is above 40%. We then randomly split the clusters into training, validation, and test set with 8:1:1 ratio. We repeat the same procedure for creating CDR-H1 and CDR-H2 splits. In total, there are 1266, 1564, and 2325 clusters for CDR-H1, H2, and H3. The size of training, validation, and test sets for each CDR is shown in the appendix.

**Metrics.** For each method, we report the perplexity (PPL) of test sequences and the root mean square deviation (RMSD) between a predicted structure and its ground truth structure reported in SAbDab. RMSD is calculated by the Kabsch algorithm (Kabsch, 1976) based on $C_\alpha$ coordinate of CDR residues. Since the mapping between sequences and structures is deterministic in RefineGNN, we can calculate perplexity in the same way as standard sequence models.

**Results.** Since the LSTM baseline does not involve structure prediction, we report RMSD for graph-based methods only. As shown in Table 1, RefineGNN significantly outperforms all baselines in both metrics. For CDR-H3, our model gives 13% PPL reduction (8.54 v.s. 9.70) over sequence only model and 10% PPL reduction over AR-GNN (8.54 v.s. 9.44). RefineGNN also predicts the structure more accurately, with 30% relative RMSD reduction over AR-GNN. In Figure 3, we provide examples of predicted 3D structures of CDR-H3 loops.

**Ablation studies.** We further conduct ablation experiments on the CDR-H3 generation task to study the importance of different modeling choices. First, when we remove the attention mechanism and context coarsening step in section 3.3, the PPL increases from 8.54 to 8.86 (Figure 3C, row 2) and 9.01 (Figure 3C, row 3) respectively. We also tried to remove both the attention and coarsening

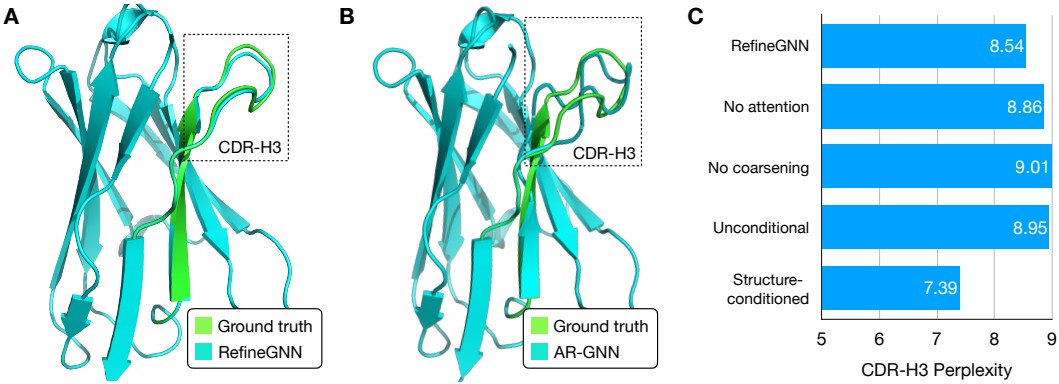

Figure 3: (A) CDR-H3 structure predicted by RefineGNN (PDB: 4bkl, RMSD = 0.57). The predicted structure (cyan) is aligned to the true structure (green) using the Kabsch algorithm. (B) CDR-H3 structure predicted by AR-GNN (PDB: 4bkl, RMSD = 2.16). (C) Ablation studies of different modeling choices in RefineGNN in the CDR-H3 perplexity evaluation task.

modules and trained the model without conditioning on the context sequence. The PPL of this unconditional variant is much worse than our conditional model (Figure 3C, row 4). Lastly, we train a structure-conditioned model by feeding the ground truth structure to RefineGNN at every generation step (Figure 3C, row 5). While this structure-conditioned model gives a lower PPL as expected (7.39 v.s. 8.54), it is not too far away from the sequence only model (PPL = 9.70). This suggests that RefineGNN is able to extract a decent amount of information from the partial structure co-evolving with the sequence.

## 4.2 Antigen-binding antibody design

**Data.** Adolf-Bryfogle et al. (2018) selected 60 antibody-antigen complexes as an antibody design benchmark. Given the framework of an antibody, the goal is to design its CDR-H3 that binds to its corresponding antigen. For simplicity, none of the methods is conditioned on the antigen structure during CDR-H3 generation. We leave antigen-conditioned CDR generation for future work.

**Metric.** Following Adolf-Bryfogle et al. (2018), we use amino acid recovery (AAR) as the evaluation metric. For any generated sequence, we define its AAR as the percentage of residues having the same amino acid as the corresponding residue in the original antibody.

**Results.** For LSTM, AR-GNN, and RefineGNN, the training set in this setup is the entire SAbDab except antibodies in the same cluster as any of the test antibodies. At test time, we generate 10000 CDR-H3 sequences for each antibody and select the top 100 candidates with the lowest perplexity. For simplicity, all methods are configured to generate CDRs of the same length as the original CDR. As shown in Table 1, our model achieves the highest AAR score, with around 7% absolute improvement over the best baseline. In Figure 4A, we show an example of a generated CDR-H3 sequence and highlight residues that are different from the original antibody. We also found that sequences with lower perplexity tend to have a lower AA recovery error (Pearson R = 0.427, Figure 4B). This suggests that we can use perplexity as the ranking criterion for antibody design.

## 4.3 SARS-CoV-2 neutralization optimization

**Data.** The Coronavirus Antibody Database (CoVAbDab) contains 2411 antibodies, each associated with multiple binary labels indicating whether it neutralizes a coronavirus (SARS-CoV-1 or SARS-CoV-2) at a certain epitope. Similar to the previous experiment, we divide the antibodies into training, validation, and test sets based on CDR-H3 cluster split with 8:1:1 ratio.

**Neutralization predictor.** The predictor takes as input the VH sequence of an antibody and outputs a neutralization probability for the SARS-CoV-1 and SARS-CoV-2 viruses. Each residue is embedded into a 64 dimensional vector, which is fed to a SRU encoder (Lei, 2021) followed by average-pooling and a two-layer feed forward network. The final outputs are the probabilities $p_1$

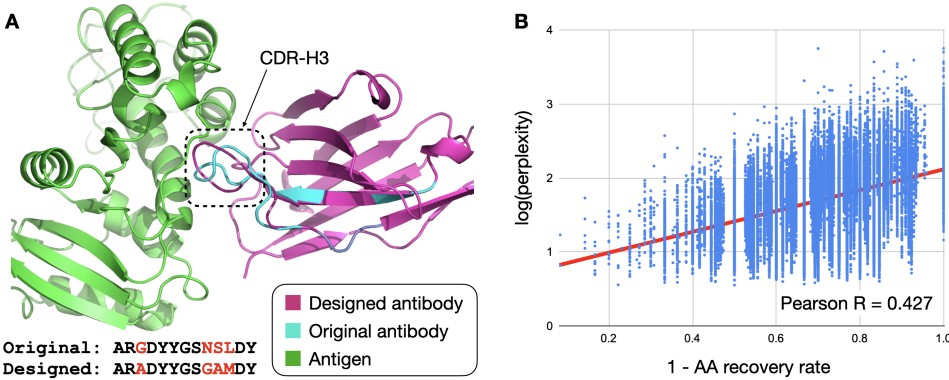

Figure 4: (A) Visualization of a generated CDR-H3 sequence and its structure in complex with an antigen (PDB: 4cmh). The predicted structure is aligned and grafted onto the original antibody using the Kabsch algorithm. Residues different from the original antibody are highlighted in red. (B) The correlation between the perplexity of a generated sequence and AA recovery error.

Table 2: SARS-CoV-2 neutralization optimization results. For each method, we report the PPL on CoVAbDab after pretraining on SAbDab and then report the average neutralization score after ITA finetuning. The average neutralization probability of original CoVAbDab antibodies is 69.3%.

|  | Original | LSTM | AR-GNN | RefineGNN |
|---|---|---|---|---|
| CoVAbDab PPL ($\downarrow$) | - | 9.40 | 8.67 | **7.86** |
| Neutralization ($\uparrow$) | 69.3% | 72.0% | 70.4% | **75.2%** |

and $p_2$ of neutralizing SARS-CoV-1 and SARS-CoV-2 and our scoring function is $f(s) = p_2$. The predictor achieved 0.81 test AUROC for SARS-CoV-2 neutralization prediction.

**CDR sequence constraints.** Therapeutic antibodies must be free from developability issues such as glycosylation and high charges (Raybould et al., 2019). Thus, we include four constraints on a CDR-H3 sequence $s$: 1) Its net charge must be between -2.0 and 2.0 (Raybould et al., 2019). The definition of net charge is given in the appendix. 2) It must not contain the N-X-S/T motif which is prone to glycosylation. 3) Any amino acid should not repeat more than five times (e.g. SSSSS). 4) Perplexity of a generated sequence given by LSTM, AR-GNN, and RefineGNN should be all less than 10. The last two constraints force generated sequences to be realistic. We use all three models in the perplexity constraint to ensure a fair comparison for all methods.

**Metric.** For each antibody in the test set, we generate 100 new CDR-H3 sequences, concatenate them with its context sequence to form 100 full VH sequences, and feed them into the neutralization predictor $f$. We report the average neutralization score of antibodies in the test set. Neutralization score of a generated sequence $s'$ equals $f(s')$ if it satisfies all the CDR sequence constraints. Otherwise the score is the same as the original sequence. In addition, we pretrain each model on the SAbDab CDR-H3 sequences and evaluate its PPL on the CoVAbDab CDR-H3 sequences.

**Results.** All methods are pretrained on SAbDab antibodies and finetuned on CoVAbDab using the ITA algorithm to generate neutralizing antibodies. Our model outperforms the best baseline by a 3% increase in terms of average neutralization score (Table 2). Our pretrained RefineGNN also achieves a much lower perplexity on CoVAbDab antibodies (7.86 v.s. 8.67). Examples of generated CDR-H3 sequences and their predicted neutralization scores are shown in the appendix.

## 5   CONCLUSION

In this paper, we developed a RefineGNN model for antibody sequence and structure co-design. The advantage of our model over previous graph generation methods is its ability to revise a generated subgraph to accommodate addition of new residues. Our approach significantly outperforms sequence-based and graph-based approaches on three antibody generation tasks.

ACKNOWLEDGEMENT

We would like to thank Rachel Wu, Xiang Fu, Jason Yim, and Peter Mikhael for their valuable feedback on the manuscript. We also want to thank Nitan Shalon, Nicholas Webb, Jae Hyeon Lee, Qiu Yu, and Galit Alter for their suggestions on method development. We are grateful for the generous support of Mark and Lisa Schwartz, funding in a form of research grant from Sanofi, Defense Threat Reduction Agency (DTRA), C3.ai Digital Transformation Institute, Eric and Wendy Schmidt Center at the Broad Institute, Abdul Latif Jameel Clinic for Machine Learning in Health, DTRA Discovery of Medical Countermeasures Against New and Emerging (DOMANE) threats program, and DARPA Accelerated Molecular Discovery program.

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

## A  MODEL ARCHITECTURE DETAILS

### A.1  REFINEGNN

**Node features.** Each node feature $\boldsymbol{v}_i$ encodes three dihedral angles as follows.

$$\boldsymbol{v}_i = (\cos\phi_i, \cos\psi_i, \cos\omega_i, \sin\phi_i, \sin\psi_i, \sin\omega_i) \tag{16}$$

**Edge features.** The orientation matrix $\boldsymbol{O}_i = [\boldsymbol{b}_i, \boldsymbol{n}_i, \boldsymbol{b}_i \times \boldsymbol{n}_i]$ defines a local coordinate system for each residue $i$ (Ingraham et al., 2019), which is calculated as

$$\boldsymbol{u}_i = \frac{\boldsymbol{x}_i - \boldsymbol{x}_{i-1}}{\|\boldsymbol{x}_i - \boldsymbol{x}_{i-1}\|}, \quad \boldsymbol{b}_i = \frac{\boldsymbol{u}_i - \boldsymbol{u}_{i+1}}{\|\boldsymbol{u}_i - \boldsymbol{u}_{i+1}\|}, \quad \boldsymbol{n}_i = \frac{\boldsymbol{u}_i \times \boldsymbol{u}_{i+1}}{\|\boldsymbol{u}_i \times \boldsymbol{u}_{i+1}\|} \tag{17}$$

**Attention mechanism.** The attention layer used in Eq.(13) is a standard bilinear attention:

$$\text{attention}(\boldsymbol{c}_{1:n}, \boldsymbol{h}_t) = \sum_i \alpha_{i,t} \boldsymbol{c}_i, \quad \alpha_{i,t} = \frac{\exp(\boldsymbol{c}_i^\top \boldsymbol{W} \boldsymbol{h}_t)}{\sum_j \exp(\boldsymbol{c}_j^\top \boldsymbol{W} \boldsymbol{h}_t)} \tag{18}$$

### A.2  AR-GNN

AR-GNN generates an antibody graph autoregressively. In each generation step $t$, AR-GNN learns to encode the current subgraph $\mathcal{G}_{1:t}$ induced from residues $\{\boldsymbol{s}_1, \cdots, \boldsymbol{s}_t\}$ into a list of vectors

$$\{\boldsymbol{h}_1, \cdots, \boldsymbol{h}_t\} = \text{MPN}_\theta(\mathcal{G}_{1:t}). \tag{19}$$

For fair comparison, we use the same MPN architecture for both RefineGNN and AR-GNN. In terms of structure prediction, AR-GNN first predicts the node feature $\hat{\boldsymbol{v}}_{t+1}$ of the next residue $t+1$, namely the dihedral angle between its three atoms $C_\alpha, C, N$.

$$\hat{\boldsymbol{v}}_{t+1} = \boldsymbol{W}_v \boldsymbol{h}_t \tag{20}$$

In addition, AR-GNN predicts the pairwise distance between $\boldsymbol{s}_{t+1}$ and previous residues $\boldsymbol{s}_1, \cdots, \boldsymbol{s}_t$.

$$\hat{\boldsymbol{d}}_{i,t+1} = \text{FFN}(\boldsymbol{W}_d \boldsymbol{h}_i + \boldsymbol{U}_d \boldsymbol{h}_t + \boldsymbol{V}_d E_{pos}(t+1-i)), \tag{21}$$

where FFN is a feed-forward network with one hidden layer and $E_{pos}$ is the positional encoding of $t+1-i$, the *gap* between residue $\boldsymbol{s}_{t+1}$ and $\boldsymbol{s}_i$ in the sequence. Lastly, AR-GNN predicts the amino acid type of residue $\boldsymbol{s}_{t+1}$ by

$$\hat{\boldsymbol{p}}_{t+1} = \text{softmax}(\boldsymbol{W}_a \boldsymbol{g}_{t+1}), \quad \{\boldsymbol{g}_1, \cdots, \boldsymbol{g}_{t+1}\} = \text{MPN}_{\theta'}(\mathcal{G}_{1:t+1}) \tag{22}$$

Note that AR-GNN also uses two separate MPNs for structure and sequence prediction. However, unlike RefineGNN, AR-GNN is trained under teacher forcing — we need to feed it the ground truth structure and sequence in each generation step. In particular, we find data augmentation to be crucial for AR-GNN performance. Data augmentation is essential because of the discrepancy between training and testing. The model is trained under teacher forcing, but it needs to decode a graph without teacher forcing at test time. We find mistakes made in previous steps have a great impact on subsequent predictions during decoding.

Specifically, for every antibody $\boldsymbol{s}$, we create a *corrupted* graph $\widetilde{\mathcal{G}}$ by adding independent random Gaussian noise to every coordinate: $\tilde{\boldsymbol{x}}_i = \boldsymbol{x}_i + 3\epsilon, \epsilon \sim \mathcal{N}(0, I)$. In each generation step, we apply MPN over the corrupted graph instead.

$$\{\widetilde{\boldsymbol{h}}_1, \cdots, \widetilde{\boldsymbol{h}}_t\} = \text{MPN}_\theta(\widetilde{\mathcal{G}}_{1:t}), \quad \{\widetilde{\boldsymbol{g}}_1, \cdots, \widetilde{\boldsymbol{g}}_{t+1}\} = \text{MPN}_{\theta'}(\widetilde{\mathcal{G}}_{1:t+1}) \tag{23}$$

The node and edge labels are still defined by the ground truth structure. Specifically, let $\boldsymbol{v}_t$ and $\boldsymbol{d}_{i,j}$ be the ground truth dihedral angle and pairwise distance calculated from the original, uncorrupted graph $\mathcal{G}$. AR-GNN loss function is defined as the following.

$$\mathcal{L}_{AR} = \sum_{i,j} \|\hat{\boldsymbol{d}}_{i,j} - \boldsymbol{d}_{i,j}\|^2 + \sum_t \|\hat{\boldsymbol{v}}_t - \boldsymbol{v}_t\|^2 + \mathcal{L}_{ce}(\hat{\boldsymbol{p}}_t, \boldsymbol{s}_t) \tag{24}$$

Similar to RefineGNN, AR-GNN also uses attention mechanism for conditional generation. Specifically, we concatenate the residue representations $\widetilde{\boldsymbol{h}}_t, \widetilde{\boldsymbol{g}}_t$ from MPN with context vectors learned from an attention layer.

$$\widetilde{\boldsymbol{h}}_t \leftarrow \widetilde{\boldsymbol{h}}_t \oplus \text{attention}(\boldsymbol{c}_{1:n}, \widetilde{\boldsymbol{h}}_t) \quad \widetilde{\boldsymbol{g}}_t \leftarrow \widetilde{\boldsymbol{g}}_t \oplus \text{attention}(\boldsymbol{c}_{1:n}, \widetilde{\boldsymbol{g}}_t) \tag{25}$$

Table 3: SARS-CoV-2 neutralization optimization results. Here we show examples of new CDR-H3 sequences generated by our model and their predicted neutralization improvement over original antibodies S1D7 and C694 in the CoVAbDab database.

| Antibody: | S1D7 | C694 |
|---|---|---|
| Old CDR-H3 | TRGHSDY | ARDRGYDSSGPDAFDI |
| New CDR-H3 | ARWWMDV | ARERIIIVSISAWMDV |
| Improvement | $63\% \rightarrow 73\%$ | $82\% \rightarrow 91\%$ |

## B  EXPERIMENTAL DETAILS

**Hyperparameters.** For AR-GNN and RefineGNN, we tried hidden dimension $d_h \in \{128, 256\}$ and number of message passing layers $L \in \{1, 2, 3, 4, 5\}$. We found $d_h = 256, L = 4$ worked the best for RefineGNN and $d_h = 256, L = 3$ worked the best for AR-GNN. For LSTM, we tried $d_h \in \{128, 256, 512, 1024\}$. We found $d_h = 256$ worked the best. All models are trained by an Adam optimizer with a dropout of 0.1 and a learning rate of 0.001.

**SAbDab data.** The dataset statistics of SAbDab is the following (after deduplication). For CDR-H1, the train/validation/test size is 4050, 359, and 326. For CDR-H2, the train/validation/test size is 3876, 483, and 376. For CDR-H3, the train/validation/test size is 3896, 403, and 437.

Since SAbDab includes both bound and unbound structures, we removed all antigens and used the bound antibody structure for training. Specifically, 65% of our training data came from bound state structures. We included all data in our training set because the mismatch between bound and unbound structures is relatively small. In fact, Al Qaraghuli et al. (2020) studied eight antibodies and found that the RMSD between bound and unbound structures over VH domains is less than 0.7 on average.

**RAbD configuration.** We provided details of the *de novo* design setup of RosettaAntibodyDesign (RAbD) here. For each antibody in the test set, RAbD starts by randomly selecting a CDR from RAbD's internal database of known CDR structures. The chosen CDR-H3 sequence is required to have same length as the original sequence, but it cannot be exactly the same as the original CDR-H3 sequence. After the initial CDR structure is chosen, RAbD grafts it onto the antibody and performs energy minimization to stabilize its structure. Next, RAbD runs 100 iterations of sequence design to modify the grafted CDR-H3 structure by randomly substituting amino acids. In each sequence design iteration, it performs energy minimization to adjust the structure according to the changed amino acid. Lastly, the model returns the generated CDR-H3 sequence with the lowest energy.

**SARS-CoV-2 neutralization.** Each generative model is pretrained on the SAbDab data to learn a prior distribution over CDR-H3 structures. Given a fixed predictor $f$, we use the ITA algorithm to finetune our pretrained models to generate neutralizing antibodies. Each model is trained for 3000 ITA steps with $M = 100$. Generated CDR-H3 sequences from our model are visualized in Table 3.

Our neutralization predictor $f$ is trained on the CoVAbDab database. For simplicity, we only consider two viruses, SARS-CoV-1 and SARS-CoV-2 since other coronavirus have very little training data. For the same reason, we only consider the spike protein receptor binding domain as our target epitope. The predictor is trained in a multi-task fashion to predict both SARS-CoV-1 and SARS-CoV-2 neutralization labels. The SRU encoder has a hidden dimension of 256. The model was trained with a dropout of 0.2, a learning rate of 0.0005, and batch size of 16.

The charge of a residue is defined as $C(\boldsymbol{s}_i) = \mathbb{I}[\boldsymbol{s}_i \in \{R, K\}] + 0.1 \cdot \mathbb{I}[\boldsymbol{s}_i = H] - \mathbb{I}[\boldsymbol{s}_i \in \{D, E\}]$ (Raybould et al., 2019). The net charge of a sequence $\boldsymbol{s}_1 \cdots \boldsymbol{s}_n$ is defined as $\sum_i C(\boldsymbol{s}_i)$.

