# OpenReview forum: "Iterative Refinement Graph Neural Network for Antibody Sequence-Structure Co-design"
_ICLR.cc/2022/Conference — ICLR 2022 Spotlight_

### Official Review · Reviewer_meMq · 2021-11-01

**Correctness:** 3
**Technical Novelty And Significance:** 3
**Empirical Novelty And Significance:** 3
**Recommendation:** 8
**Confidence:** 3

**Main Review:**

Strengths:
- The paper is well-written and easy to follow.
- The proposed refinement method has high novelty and outperform state-of-the-art baseline methods.

Weakness:
- The experimental evaluation may be problematic, it is not convincing to use machine learning methods to predict the neutralization ability based on CDR H3.

**Summary Of The Paper:**

The paper proposed a deep generative model named iterative refinement graph neural network to generate antibody CDR for Y-shaped antibodies. Specifically, it sequentially generates the CDR residue sequence and refines the global structure iteratively. Empirical results show superior performance compared with baselines.

**Summary Of The Review:**

The proposed method is novel and the paper is well written, validated by thorough empirical studies.

---

> ### Author Response · Authors · 2021-11-23
> **Thank you for your review**
>
> Dear reviewer,
>
> Thank you for your insightful comments and positive review!
>
> Q1: The experimental evaluation may be problematic, it is not convincing to use machine learning methods to predict the neutralization ability based on CDR H3.
>
> To clarify, the neutralization predictor is based on the full heavy chain variable domain (VH) sequence rather than CDR-H3 alone. For a given antibody, we use our generative model to modify its CDR-H3 sequence, concatenate the new CDR-H3 with the original framework region to get a full VH sequence, and feed the full sequence into the predictor.
>
> In small molecule drug design, it is a common practice to use predictors to evaluate designed molecules. Since this practice is still new to the antibody engineering community, section 4.3 is only a proof of concept for predictor-based neutralization optimization and we are currently validating the designed antibodies experimentally.

---

### Official Review · Reviewer_9GqB · 2021-11-03

**Correctness:** 3
**Technical Novelty And Significance:** 4
**Empirical Novelty And Significance:** 4
**Recommendation:** 8
**Confidence:** 4

**Main Review:**

This paper studies the important problem of computationally designing antibody CDRs. The joint modeling approach of structures and sequences is novel and interesting. The generation method is flexible and the authors also adapt it for conditional generation (conditioned on the rest of the antibody and on given properties of interest with the predictive model already available).

The empirical results are convincing. Both perplexity and sequence recovery are standard metrics in protein design, and the proposed method showed improvements in both perplexity and sequence recovery on two separate data sets. The two baselines used for comparison are well described and meaningful.

Request for clarification: In section 4.2, it is not immediately clear whether the antigen is also included in the conditioning or only the antibody itself.

It would be appreciated if the authors can add more discussion around when this proposed method applies in practice. For example, the method requires already knowing the frame of the antibody. When is a fixed frame variable CDR-H3 design a reasonable assumption? Or, when can we realistically expect to have accurate predictors for properties of interest?

While the model is trained on unbound antibody structures, the second evaluation task is conditioned on the antibody structures in bound state within a complex (if I’m not mistaken), will there be a mismatch between the bound and unbound structures? Clarification and discussion around this point would also be appreciated.


**Summary Of The Paper:**

This paper proposes a joint generative model (co-designing the sequence and the structure at the same time) for the CDRs of antibodies, with the goal to enhance binding specificity or neutralization capabilities.

The proposed method and two existing baselines are evaluated on 1) perplexity on hold-out set; 2) perplexity and sequence recovery on known antigen-antibody complexes; and 3) redesign CDR-H3 of existing antibodies for coronavirus neutralization as measured by a given neutralization predictor. RefineGNN, the proposed method, shows improved performance in all three tasks.

**Summary Of The Review:**

Interesting approach to an important problem. Convincing empirical results and baselines. The approach might be limited to specific use cases in practice depending on the availability of a predictive model for the properties of interest and the knowledge of the antibody frame.

---

> ### Author Response · Authors · 2021-11-23
> **Thank you for your review**
>
> Dear Reviewer,
>
> Thank you for your insightful comments and positive review!
>
> Q1: In section 4.2, it is not immediately clear whether the antigen is also included in the conditioning or only the antibody itself.
>
> The antigen is not included in our model nor the baselines. We have clarified this point in section 4.2. Adding antigen structure is certainly important and it will be our future work.
>
> Q2: The method requires already knowing the frame of the antibody. When is a fixed frame variable CDR-H3 design a reasonable assumption?
>
> CDR-H3 design with a fixed framework is broadly adopted in the antibody engineering community, including physics-based methods like RosettaAntibodyDesign (Adolf-Bryfogle et al. 2018) and deep learning approaches (Liu et al., 2020). This assumption is reasonable since many antigen contacts are located in CDR-H3. Nevertheless, we agree that co-designing the framework region is also valuable and we plan to implement it in our future work.
>
> Q3: When can we realistically expect to have accurate predictors for properties of interest?
>
> Predictor accuracy depends on the size, quality, and diversity of its training data. In this paper, the SARS-CoV-2 neutralization predictor achieves around 0.81 test AUROC. We also trained a neutralization predictor for SARS-CoV-1 using the data from CoVAbDab and it achieved 0.92 test AUROC.
>
> Q4: While the model is trained on unbound antibody structures, the second evaluation task is conditioned on the antibody structures in a bound state within a complex (if I’m not mistaken). Will there be a mismatch between the bound and unbound structures?
>
> To clarify, our data from the Structured Antibody Database (SAbDab) includes both bound and unbound structures. If an antibody is in a complex, we remove all antigens and use the bound antibody structure for training. In fact, 65% of our training data come from bound state structures. We decided to include all data in our training set due to the small size of SAbDab.
>
> The mismatch between bound and unbound structures is usually small. Al Qaraghuli et al. studied eight antibodies and found that the RMSD between bound and unbound structures (variable domain residues) is less than 0.7 on average.
>
> References
>
> [1] Al Qaraghuli, M.M., Kubiak-Ossowska, K., Ferro, V.A. et al. Antibody-protein binding and conformational changes: identifying allosteric signalling pathways to engineer a better effector response. Sci Rep 10, 13696 (2020). https://doi.org/10.1038/s41598-020-70680-0
>
> [2] Jared Adolf-Bryfogle, Oleks Kalyuzhniy, Michael Kubitz, Brian D Weitzner, Xiaozhen Hu, YumikoAdachi, William R Schief, and Roland L Dunbrack Jr.  RosettaAntibodyDesign (rabd): A general framework for computational antibody design. PLoS computational biology,  14(4):e1006112,2018
>
> [3] Ge Liu, Haoyang Zeng, Jonas Mueller, Brandon Carter, Ziheng Wang, Jonas Schilz, Geraldine Horny,  Michael E Birnbaum, Stefan Ewert, and David K Gifford. Antibody complementarity determining region design using high-capacity machine learning. Bioinformatics, 36(7):2126–2133, 2020

---

### Official Review · Reviewer_NDVA · 2021-11-04

**Correctness:** 4
**Technical Novelty And Significance:** 3
**Empirical Novelty And Significance:** 4
**Recommendation:** 8
**Confidence:** 4

**Main Review:**

This paper provides an interesting study of antibody loop generation with both novel methodology and extensive empirical evaluation. This combination of strengths makes it an excellent paper that should be of wide interest.

Strengths:
- The design of each component follows naturally from previous work (such as Ingraham et al 2019). This helps isolate out key methodological contributions, for example no teacher forcing in structure, in an easy-to-interpret way.
- The coarsening procedure is of wider applicability in protein modeling beyond sampling CDR loops and an ablation study demonstrates its value empirically for this work.
- Baselines chosen highlight key interactions in RefineGNN components and show that the model appears to effectively leverage the added complexity.
- Evaluations are thorough and span a number of datasets, one of the biggest strengths of this paper.

Weaknesses:
- The use of a predictor to evaluate neutralization is justified based on very recent work, and it is unclear that this practice is in line with broader norms in the antibody engineering community.


**Summary Of The Paper:**

This paper introduces a method for jointly sampling structure and sequence of antibody loops in an iterative fashion that constrains structure generation less.

**Summary Of The Review:**

This paper provides an interesting study of antibody loop generation with both novel methodology and extensive empirical evaluation. This combination of strengths makes it an excellent paper that should be of wide interest.

---

> ### Author Response · Authors · 2021-11-23
> **Thank you for your review**
>
> Dear reviewer,
>
> Thank you for your insightful comments and positive review!
>
> Q1: The use of a predictor to evaluate neutralization is justified based on very recent work, and it is unclear that this practice is in line with broader norms in the antibody engineering community.
>
> The SARS-CoV-2 neutralization optimization task is a synthetic evaluation due to the use of a neural network predictor. This practice is commonly used in small molecule drug design, but still new to the antibody engineering community. This task is only a proof of concept for predictor-based antibody optimization and we are currently validating the designed antibodies experimentally

---

> > ### Comment · Reviewer_NDVA · 2021-11-29
> > **Thank you!**
> >
> > Thanks for your response. I look forward to seeing future in-vitro and in-vivo results from these models.

---

### Decision · Program_Chairs · 2022-01-20

**Decision:**

Accept (Spotlight)

**Comment:**

This paper proposes use of a novel generative modelling approach, over both sequences and structure of proteins, to co-design the CDR region of antibodies so achieve good binding/neutralization. The reviewers are in agreement that the problem is one of importance, and that the technical and empirical contributions are strong. There are concerns over the relevance of evaluating the method by using a predictive model as ground truth. Still, the overall contributions remain.